# How not to Lie with a Benchmark: Rearranging NLP Learderboards

## Abstract

Comparison with a human is an essential requirement for a benchmark for it to be a reliable measurement of model capabilities. Nevertheless, the methods for model comparison could have a fundamental flaw - the arithmetic mean of separate metrics is used for all tasks of different complexity, different size of test and training sets.

In this paper, we examine popular NLP benchmarks' overall scoring methods and rearrange the models by geometric and harmonic mean (appropriate for averaging rates) according to their reported results. We analyze several popular benchmarks including GLUE, SuperGLUE, XGLUE, and XTREME. The analysis shows that e.g. human level on SuperGLUE is still not reached, and there is still room for improvement for the current models.

## 1   Introduction

The benchmarking approach has a rich history throughout computer science and is now the leading method in machine learning progress validation. In the field of Natural Language Processing (NLP), there exist at least 898 benchmarks[1], the most prominent being GLUE, SuperGLUE, XGLUE, etc., created within a single paradigm.

The increase in the number of publications and developments in the field of machine learning has led to the need for methodological development of standards for describing models and all stages of the experiment, including the collection and processing of preliminary data for training, reproducibility of results, testing conditions, and most importantly, the creation of common measurable criteria for evaluating intelligent systems, both natural and artificial (Chollet, 2019). Modern NLP benchmarks are substantively inherit to Turing test, i.e. test the model abilities with various intellectual tasks expressed with texts, and methodologically inherit the benchmark approach for measuring the computing performance, like SPEC[2].

The delicate question of a general assessment of the model results on all the tasks is often solved by a method that is unforgivably simple for such a responsible task - the arithmetic mean for all the tasks. This does not take into account the scatter of results on different tasks, the different size of the task test sets (e.g. in SuperGLUE they differ a hundred times, compare 146 test samples in Winograd Schema and 10'000 test samples in ReCoRd(Wang et al., 2019a)), different susceptibility to leaks (Elangovan et al., 2021), including year of creation (Recognizing Textual Entailment data was collected in 2005 (Dagan et al., 2005), while BoolQ or CommitmentBank data was collected in 2019(Clark et al., 2019), (De Marneffe et al., 2019)).

---

[1]according to https://paperswithcode.com/area/natural-language-processing
[2]https://www.spec.org/benchmarks.html

Submitted to 35th Conference on Neural Information Processing Systems (NeurIPS 2021). Do not distribute.

In this article we present an analysis of the NLP benchmarks' results, using not the arithmetic mean, but the other metrics: geometric mean and harmonic mean. As F1 (harmonic mean) is frequently used to normalize Precision and Recall as they are fractions, optimizing the classifier threshold to maximize F1 leads to a more balanced balance between metrics than the arithmetic mean because it penalizes systems more for the smaller values(Sasaki et al., 2007). The geometric mean, as noted in the (Fleming and Wallace, 1986), is the preferred metric to the arithmetic mean in computing performance benchmarks when it comes to normalized values and percentages. The results change the usual idea of the models' order on leaderboards: humans still occupy the first place in the intellectual task solving, and the best results (1.5-2% worse than humans) belong to DeBerta(He et al., 2020), T5+Meena (Raffel et al., 2020) and McAlbert+DKM models[3]. Thus, the contribution of this paper is two-fold: 1) we present the reviewed approach model evaluation on multiple tasks 2) we re-arrange the currently existing leaderboards of most popular benchmarks.

This work is organized as follows: section 2 presents previous work on the topic, it is followed by section 3 with a description of the general methodology of cross-checking the results, including the scores for each benchmark. Next, an analysis of the results and discussion are presented in section 4, as well as a conclusion in section 5.

## 2 Previous Work

Evaluation and comparison of NLP models beget a rich history, rising with the Turing test (Turing, 2009). The question-answering approach then evolved into the SQuAD task (Rajpurkar et al., 2016), comparing systems and annotator results by their ability to find answers to informative questions. The next step in the development and assessment of intelligent systems belongs to the benchmark methodology, which aims to bring the solution of the Natural Language Understanding problem closer - General Language Understanding Evaluation(Wang et al., 2019b).

The General Language Understanding Evaluation (GLUE) methodology includes:

1. a benchmark from N (11 in original GLUE) intellectual tasks of understanding a natural language, with a fixed division into training, validation and test data;

2. a set of diagnostic data designed exclusively for testing and analyzing the results of trained systems in relation to a wide range of categories found at various levels of natural language (morphological, lexical, syntactic, semantic);

3. averaged human performance evaluation on the tasks;

4. publicly available rating system and codebase to quickly reproduce results from publicly available systems and self-evaluate models.

GLUE has developed, by no means, a prolific method to model evaluation, and has already been reproduced several times in new language material: in Chinese (Xu et al., 2020), Korean (Park et al., 2021), Russian (Shavrina et al., 2020), Polish (Rybak et al., 2020), and French (Le et al., 2019) languages, and also jumpstarted two multilingual projects: XGLUE (Liang et al., 2020) and XTREME (Hu et al., 2020).

As stated in (Wang et al., 2019a), *"Lacking a fair criterion with which to weight the contributions of each task to the overall score, we opt for the simple approach of weighing each task equally, and for tasks with multiple metrics, first averaging those metrics to get a task score."* All the GLUE-based benchmarks follow this methodology.

However, apart from the GLUE format, other benchmarks have provided several alternatives to evaluate the overall model contribution.

KILT, a Benchmark for Knowledge Intensive Language Tasks (Petroni et al., 2020), avoids calculating the overall result, and also do not compare the result with the human level, but only provides metrics for individual tasks.

DecaNLP (McCann et al., 2018) makes a rating using not the average, but the sum of points for all tasks. This approach allows balancing the contributions of different tasks to the overall metric.

---

[3]`https://www.iflytek.com/news/2118`

# 3 Method

We arrange the NLP benchmark results using the publicly available model scores for all the tasks to calculate new overall scores.

Other available options from Pythagorean means - harmonic mean and geometric mean - can also be considered: we have centered our research around 2 simple statistics that are widely used for averaging fractions (lun Chou, 1969) or normalized values (Fleming and Wallace, 1986) among the possible alternatives. We did not consider other measures of central tendency, like median and mode, as the averaged samples more often consist of about 10 measurements, and on them such metrics can give the same results for competing systems.

- The arithmetic mean (AM) desribed in eq. 1 is calculated as the sum of the task scores (Xs) divided by the total number of tasks, referred to as N.
- The geometric mean (GM) desribed in eq. 2 is calculated as the N-th root of the product of all task scores (with the above conditions), where N is the number of values.
- The harmonic mean (HM) desribed in eq. 3 is calculated as the number of values N divided by the sum of the reciprocal of the values

$$AM = \frac{1}{n}\sum_{i=1}^{n} x_i = \frac{x_1 + x_2 + \cdots + x_n}{n} \tag{1}$$

$$GM = \left(\prod_{i=1}^{n} x_i\right)^{\frac{1}{n}} = \sqrt[n]{x_1 x_2 \cdots x_n} \tag{2}$$

$$HM = \frac{N}{\frac{1}{x_1} + \frac{1}{x_2} + \cdots + \frac{1}{x_N}} \tag{3}$$

The sections below present the results of the leaderboard re-weighting as of May 2021.

## 3.1 Reevaluating the Benchmarks

The GLUE *(11 tasks for English)*, SuperGLUE *(10 tasks for English)*, XGLUE *(11 tasks for 19 languages)*, and XTREME *(4 tasks for 40 languages)* provide different scoring metrics for each task, including Accuracy, F1, Matthew's correlation coefficient, Exact Match, while the overall score is calculated by their simple average. In cases like these, the geometric mean is appropriate when the data contains values with different units of measure (lun Chou, 1969).

The harmonic mean of the task results as a better overall metric has the same grounding as introduction of the F-1 measure over precision and recall (Sasaki et al., 2007): the harmonic mean is more intuitive than the arithmetic mean when computing a mean of ratios. Given the set of metrics with a large scatter, the harmonic mean will be less than the arithmetic mean, penalizing the system more for the errors made.

As stated in (Dittmann and Maug, 2008), *"error measures are inherently subjective as they are determined by the loss function of the researcher or analyst who needs to choose a valuation procedure. Therefore, our analysis cannot establish which error measure should be used. Instead, our objective is to highlight the effects of the choice of error measure, so that researchers and analysts alike can draw their own conclusions about the error measure and, eventually, about the valuation methods they wish to use."* Nevertheless, the arithmetic, geometric and harmonic mean are all differently subjected to outliers in a data sample. The arithmetic mean always results in a higher values than the geometric mean or the harmonic mean, and the harmonic mean always results in a lower estimate than the geometric mean (Xia et al., 1999), thus, the harmonic and geometric mean tend more strongly toward the least values, and tend to mitigate the impact of large outliers and aggravate the impact of small ones.

The harmonic mean is the appropriate mean if the data is comprised of rates, while the geometric mean is used as an unbiased estimation when working with normalized ratios, for example, in finance (Dittmann and Maug, 2008) or computing benchmarks (Fleming and Wallace, 1986).

However, their applicability to a better summarization of the model performance to a single number has been widely discussed, see (Smith, 1988), discussing performance computing:

- the **harmonic mean** is considered the appropriate metric to summarize benchmark results expressed as rates,
- while **geometric mean** is applicable in case of the use of performance numbers that are normalized with respect to one of the results being compared (see 4),
- and **arithmetic mean** should not be used as a summarizing metric with rates, making it the worst choice for results accumulation.

When measuring the geometric and harmonic mean, the following assumptions were used:

- as the geometric mean does not accept zero values (also negative ones), we filled the cases of metric lacking (for example, Sentence Retrieval in XTREME) with 0.00001 values;
- tasks with more than one metric measured (e.g. Accuracy and F1), are subjected to the averaging operation when measuring the total score, as in the standard GLUE methodology: the arithmetic mean of all metrics is taken for each task. Another modification of measurement is potentially possible: for tasks with several metrics, take all of them at once and add them as separate independent results to the averaging. Then tasks with separate high metrics will have less weight on the total;
- among the best benchmark results, there were no negative values (MCC metric), but in theory they could be, and that would prevent the calculation of the harmonic mean.
- the GLUE diagnostic dataset does not have a human evaluation score on the leaderboard and therefore, is considered 0 (0.00001), though in the SuperGLUE benchmark the same dataset has obtained its evaluation.

## 3.2 GLUE

GLUE benchmark (Wang et al., 2019b) combines 11 tasks in various text classification and question answering.

**Overall score:** average of all the task results. If task has 2 main metrics, these metrics are averaged, then added to the overall average.

**Human evaluation:** collected on reported human performance numbers from original datasets, not exceeding 200 examples (heavily criticised in (Nangia and Bowman, 2019)). The human baseline performance on the diagnostic set was provided by the project authors with the help of six NLP researchers annotating 50 randomly selected sentence pairs.

**Rearranging the scores:** the results of geometric and harmonic mean rearrangement are presented in Tab. 1. GLUE benchmark seem to be the most reordered of all the ratings considered: the best result by geometric and harmonic means belongs to humans, DeBerta and McAlbert+DKM got a 1 point demotion, and the other models got severely rearranged their places.

| N | Name | AM | HM | GM | CoLA | SST-2 | MRPC Mean | STS-B Mean | QQP Mean | MNLI m | MNLI mm | QNLI |
|---|---|---|---|---|---|---|---|---|---|---|---|---|
| 16 | Human | 87,10 | 86,16 | 86,91 | 66,40 | 97,80 | 83,55 | 92,65 | 69,95 | 92,00 | 92,80 | 91,20 |
| 1 | DeBERTa Mac | 90,80 | 84,78 | 86,25 | 71,50 | 97,50 | 93,00 | 92,75 | 83,50 | 91,90 | 91,60 | 99,20 |
| 2 | Albert +DKM | 90,70 | 84,70 | 86,13 | 74,80 | 97,00 | 93,55 | 92,70 | 82,65 | 91,30 | 91,10 | 97,80 |
| 6 | T5 | 90,30 | 84,48 | 85,92 | 71,60 | 97,50 | 91,60 | 92,95 | 82,85 | 92,20 | 91,90 | 96,90 |
| 4 | PING-AN | 90,60 | 84,26 | 85,83 | 73,50 | 97,20 | 93,00 | 92,70 | 83,55 | 91,60 | 91,30 | 97,50 |
| 5 | ERNIE | 90,40 | 84,27 | 85,75 | 74,40 | 97,50 | 92,45 | 92,80 | 83,05 | 91,40 | 91,00 | 96,60 |

Table 1: Top results of ranking GLUE benchmark with geometric mean. N – original model rank on the leaderboard. MNLI m and MNLI mm correspond to MultiNLI Matched MultiNLI Mismatched, other task abbreviations correspond to their GLUE leaderboard designations accordingly.

## 3.3 SuperGLUE

SuperGLUE (Wang et al., 2019a) is the sophisticated version of the GLUE benchmark, combining 10 tasks with a higher demand for higher intellectual abilities. Task data must is available under various licenses that allow use and redistribution for research purposes.

162 **Overall score:** average of all the task results. If task has 2 main metrics, these metrics are averaged,
163 then added to the overall average.

164 **Human evaluation:** ready-made estimates for WiC, MultiRC, RTE, and ReCoRD datasets, the
165 other tasks being evaluated by the project creators with the help of crowdworker annotators through
166 Amazon's Mechanical Turk.

167 **Rearranging the scores:** Re-weighting the results using the geometric mean and harmonic mean
168 again makes significant changes to the original ranking: the top-3 result (human) is ranked top-1, the
169 DeBerta and T5 models are shifted down 1 position, PAI ALbert and Nezha Plus models swap their
170 places, see Tab. 2.

| N | Model | AM | HM | GM | BoolQ | CB Mean | COPA | MRC | RCD | RTE | WiC | WSC | AX-b | AX-g mean |
|---|-------|-----|-----|-----|-------|---------|------|-----|-----|-----|-----|-----|------|-----------|
| 3 | Human | 89,80 | 87,96 | 88,73 | 89,00 | 97,35 | 100,00 | 66,85 | 91,50 | 93,60 | 80,00 | 100,00 | 76,6 | 99,5 |
| 1 | DeBERTa | 90,30 | 86,89 | 87,60 | 90,40 | 96,65 | 98,40 | 75,95 | 94,30 | 93,20 | 77,50 | 95,90 | 66,7 | 93,55 |
| 2 | T5+ Meena | 90,20 | 86,42 | 87,10 | 91,30 | 96,70 | 97,40 | 75,65 | 93,85 | 92,70 | 77,90 | 95,90 | 66,5 | 89,35 |
| 4 | T5 | 89,30 | 85,89 | 86,57 | 91,20 | 95,35 | 94,80 | 75,70 | 93,75 | 92,50 | 76,90 | 93,80 | 65,6 | 92,3 |
| 6 | PAI Albert | 86,10 | 85,24 | 85,78 | 88,10 | 94,40 | 91,80 | 69,65 | 88,65 | 88,80 | 74,10 | 93,20 | 75,6 | 98,75 |
| 5 | Nezha plus | 86,70 | 81,30 | 82,29 | 87,80 | 95,20 | 93,60 | 69,85 | 89,85 | 89,10 | 74,60 | 93,20 | 58,00 | 80,75 |

Table 2: Top results of ranking SuperGLUE benchmark with geometric mean. N – original model rank on the leaderboard MRC stands for MultiRC averaged metric, RCD - ReCoRD averaged metric.

## 3.4   XTREME

172 The XTREME benchmark (Hu et al., 2020)covers 40 typologically diverse languages from 12
173 language families and includes 9 tasks that require analysis of different levels of syntax or semantics.

174 **Overall score:** 2-step averaging: 1) calculating average for each task on all languages 2) calculating
175 average on all tasks.

176 **Human evaluation:** 2-step averaging:

177   1. step 1: ready-made estimates from the original datasets taken and extrapolated to all
178      unestimated languages; besides, for some datasets there were no original estimates provided
179      (POS) and an empirical estimate of 97% was taken based on (Manning, 2011); no estimates
180      for NER and sentence retrieval tasks;

181   2. step 2: all the task results averaged together.

182 **Rearranging the scores:** the results of applying the geometric mean and harmonic mean did not
183 change the current ranking of the models - the quality spread between them is high enough for the
184 metrics averaging them to retain the current order.

| N | Model | AM | HM | GM | Sentence-pair Classification | Structured Prediction | Question Answering | Sentence Retrieval |
|---|-------|-----|-----|-----|------------------------------|-----------------------|--------------------|--------------------|
| 1 | Human | 93,30 | 93,13 | 93,21 | 95,10 | 97,00 | 87,80 | 0.00001 |
| 2 | VECO | 81,10 | 81,27 | 81,70 | 88,60 | 75,40 | 72,40 | 92,10 |
| 3 | ERNIE-M | 80,90 | 81,11 | 81,52 | 87,90 | 75,60 | 72,30 | 91,90 |
| 4 | T-ULRv2 | 80,70 | 80,91 | 81,25 | 88,80 | 75,40 | 72,90 | 89,30 |
| 5 | Anonymous3 | 79,90 | 80,12 | 80,50 | 88,20 | 74,60 | 71,70 | 89,00 |
| 6 | Polyglot | 77,80 | 78,02 | 78,56 | 87,80 | 72,90 | 67,40 | 88,30 |

Table 3: Top results of ranking XTREME benchmark with geometric mean. N – original model rank on the leaderboard; the averaged task scores are shown by the column markings.

## 3.5   XGLUE

186 The XGLUE benchmark(Liang et al., 2020)consists of 11 problems in 19 languages and evaluates
187 the performance of multilingual pre-trained systems in terms of their ability to cross-language
188 understanding and natural language generation.

**Overall score:** 2-step averaging: 1) calculating average for each task on all languages 2) calculating average on all tasks.

**Human evaluation:** not provided.

**Rearranging the scores:** Since the human level is not measured in the benchmark, we can only compare the 2 present models with each other. Tab. 4 shows the results - the difference in the quality of the models is large enough to preserve their ranking on all averaging metrics.

| N | Model | AM | HM | GM | NER | POS | NC | MLQA | XNLI | PAWS-X | QADSM | WPR | QAM |
|---|-------|------|------|------|------|------|------|------|------|--------|-------|------|------|
| 1 | FILTER | 80,10 | 79,61 | 79,86 | 82,60 | 81,60 | 83,50 | 76,20 | 83,90 | 93,80 | 71,40 | 74,70 | 73,40 |
| 2 | Unicoder Baseline | 76,10 | 75,45 | 75,80 | 79,70 | 79,60 | 83,50 | 66,00 | 75,30 | 90,10 | 68,40 | 73,90 | 68,90 |

| N | Model | AM | HM | GM | QG | NTG |
|---|-------|------|------|------|------|------|
| 1 | Unicoder Baseline | 10,70 | 10,65 | 9.10 | 10,60 | 10,70 |
| 2 | MP-Tune | 8,70 | 8,70 | 7.10 | 8,10 | 9,40 |

Table 4: Top results of ranking XGLUE benchmark with geometric mean. N – original model rank on the leaderboard; the first 3 rows correspond to NLU tasks, the last 3 rows - to the NLG tasks.

# 4 Results and Discussion

The results show that the ranking of results within a single leaderboard can fluctuate significantly. So, in GLUE, the first place in terms of the harmonic and geometric mean belongs to the result occupying the 16th line in the arithmetic mean. In SuperGLUE, the permutation is not so striking - the third result is on the 1st place. On the XTREME and XGLUE benchmarks system ranking is preserved.

Since all three averaging metrics considered are subject to different biases, we present the statistical measurements of the top-3 SuperGLUE results in Tab. 5. Human results have the highest total points for all tasks (as in the DecaNLP methodology), while the standard deviation and variance are greater than top-2 and top-3 models.

| Model | AM | GM | HM | Sum | Var | Std |
|-------|------|-------|-------|--------|--------|-------|
| Human | 89,8 | **88,73** | **87,96** | **894,40** | 130,31 | 11,42 |
| DeBerta | **90,3** | 87,60 | 86,89 | 882,55 | 117,46 | 10,84 |
| T5 + Meena | 90,2 | 87,10 | 86,42 | 877,25 | **112,39** | **10,60** |

Table 5: Measuring the statistics of the top-3 SuperGLUE results. Sum is a sum of all the task scores, Var and Std are variance and standard deviation on the task scores respectively. Notable results are highlighted in bold.

To further explore the rating results of the GLUE and SuperGLUE benchmarks, we have conducted a series of experiments with normalizing the model performance with human scores, fully transferring the standard methodology for computing performance, in which the geometric mean is the approved metric. The results are presented in Appendix 1. As can be concluded from the table with normalized values, in the case of SuperGLUE, the results obtained by the new ranking method are confirmed. In the case of GLUE, the geometric mean shows that the normalized ratio of models to the human level asserts a high level of artificial solutions over the human level.

The following topics remain debatable and need special attention of the community:

1. Different metrics for obtaining the average value (arithmetic, geometric, harmonic) have different restrictions on the accepted values (for example, not every one can take negative or zero values). At the same time, metrics that take zero and negative values are actively used in measuring various skills - MCC metric on SuperGLUE diagnostics can be negative, other metrics can be equal to or greater than zero, and they are averaged altogether. Potentially, the issue of a fair metric will raise the problem of revising the use of some individual metrics for evaluating tasks. The differences in the metric scale for different tasks can pose problems for the computation of the total score and some metrics can be scaled or normalized. We may consider rescaling MCC score so that it is between 0 and 1.

2. Correct averaging of the overall score for multilingual benchmarks creates additional problems while performing the averaging operation in 2 stages: for all languages and all tasks.

As a result, the score, consisting of one number, becomes less and less informative and more prone to outliers.

3. Nevertheless, the competitive side of benchmarks is the driving force behind the progress in the field of machine learning, and besides all problems, it is still not worth giving up leaderboards with a single metric.

4. In addition to the problem of the main averaging metric, we left outside of the scope the problem that was also discovered within the framework of this study: human benchmark scores on various tasks were obtained in a very different way, and always on a smaller sample than the full test set. For a fair comparison of humans and machines, the test results should be normalized by the same number of test items, and it is worth revising the human evaluation and re-performing it on all test items using more annotators.

## 5  Conclusion

In this paper we present an alternative methods to arrange the popular NLP benchmark results, elaborating on several task evaluation. We analyze popular benchmark averaging methods and provide new insight into model comparison. Namely, we obtain the following results:

- for popular benchmarks GLUE and SuperGLUE we can conclude that their overall score is subject to bias due to outliers; the alternative arrengemend methods end with significantly different ordering of the results;
- rebuilding leaderboards using other metrics (geometric or harmonic mean) allows one to conclude that human result is the first in the rankings;
- in XGLUE leaderboard human result is obtained by extrapolation from one language to others, while in practice the level of problem-solving by native speakers of different languages varies;
- the last finding could be extended to other multilingual benchmarks also.

An unbiased view of the overall score in the benchmarks is a necessity for a community to target language model development on the complex improvement of their quality, not the partial results in narrow tasks. The expansion towards multilingual and multimodal models makes this issue more and more urgent and we hope our help to foster research in this direction.

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
