# Appendix

See tables below

## 5.1 Appendix 1

## 5.2 Appendix 2

## 5.3 Appendix 3

Table 6: SuperGLUE benchmark results normalized to human level: geometric mean is the main metric

| Rank | SuperGLUE Model | HM | GM | AM | BoolQ | CB mean | COPA | MultiRC Mean | ReCoRD mean | RTE | WiC | WSC | AX-b | AX-g mean |
|---|---|---|---|---|---|---|---|---|---|---|---|---|---|---|
| 3 | Human | **87,9 (1)** | **88,7 (1)** | **89,8 (1)** | 89 (1) | 97,35 (1) | 100 (1) | 66,85 (1) | 91,5 (1) | 93,6 (1) | 80 (1) | 100 (1) | 76,6 (1) | 99,5 (1) |
| 1 | DeBERTa | **0,985** | **0,987** | **0,989** | 1,016 | 0,993 | 0,984 | 1,136 | 1,031 | 0,996 | 0,969 | 0,959 | 0,871 | 0,940 |
| 2 | T5 + Meena | **0,979** | **0,982** | **0,984** | 1,026 | 0,993 | 0,974 | 1,132 | 1,026 | 0,990 | 0,974 | 0,959 | 0,868 | 0,898 |
| 4 | T5 | **0,973** | **0,976** | **0,978** | 1,025 | 0,979 | 0,948 | 1,132 | 1,025 | 0,988 | 0,961 | 0,938 | 0,856 | 0,928 |

| Rank | GLUE Model | HM | GM | AM | CoLA | SST-2 | MRPC Mean | STS-B Mean | QQP Mean | MNLI-m | MNLI-mm | QNLI | RTE | WNLI |
|---|---|---|---|---|---|---|---|---|---|---|---|---|---|---|
| 16 | Human | **86,1 (1)** | **86,9 (1)** | **87,1 (1)** | 66,4 (1) | 97,8 (1) | 83,55 (1) | 92,65 (1) | 69,95 (1) | 92 (1) | 92,8 (1) | 91,2 (1) | 93,6 (1) | 95,9 (1) |
| 1 | DeBERTa | **0,995** | **1,004** | **1,012** | 1,077 | 0,997 | 1,113 | 1,001 | 1,194 | 0,999 | 0,987 | 1,088 | 0,996 | 0,985 |
| 2 | McAlbert | **0,993** | **1,002** | **1,011** | 1,127 | 0,992 | 1,120 | 1,001 | 1,182 | 0,992 | 0,982 | 1,072 | 0,983 | 0,985 |
| 6 | T5 | **0,991** | **1,000** | **1,008** | 1,078 | 0,997 | 1,096 | 1,003 | 1,184 | 1,002 | 0,990 | 1,063 | 0,991 | 0,985 |

Table 7: GLUE results with all metrics for the tasks, including tasks with double metrics.

| N | Model | GM | AM | CoLA | SST-2 | MRPC F1 | MRPC Acc | MRPC Mean | STS-B Prs | STS-B Spr | STS-B Mean | QQP F1 | QQP Acc | QQP Mean | MNLI m | MNLI mm | QNLI | RTE | WNLI |
|---|---|---|---|---|---|---|---|---|---|---|---|---|---|---|---|---|---|---|---|
| 16 | Human | **86,906** | 87,1 | 66,4 | 97,8 | 86,3 | 80,8 | 83,55 | 92,7 | 92,6 | 92,65 | 59,5 | 80,4 | 69,95 | 92 | 92,8 | 91,2 | 93,6 | 95,9 |
| 1 | DeBERTa | **86,248** | 90,8 | 71,5 | 97,5 | 94 | 92 | 93 | 92,9 | 92,6 | 92,75 | 76,2 | 90,8 | 83,5 | 91,9 | 91,6 | 99,2 | 93,2 | 94,5 |
| 2 | HFL iFLYTEK | **86,127** | 90,7 | 74,8 | 97 | 94,5 | 92,6 | 93,55 | 92,8 | 92,6 | 92,7 | 74,7 | 90,6 | 82,65 | 91,3 | 91,1 | 97,8 | 92 | 94,5 |
| 6 | T5 | **85,915** | 90,3 | 71,6 | 97,5 | 92,8 | 90,4 | 91,6 | 93,1 | 92,8 | 92,95 | 75,1 | 90,6 | 82,85 | 92,2 | 91,9 | 96,9 | 92,8 | 94,5 |
| 4 | PING-AN | **85,827** | 90,6 | 73,5 | 97,2 | 94 | 92 | 93 | 93 | 92,4 | 92,7 | 76,1 | 91 | 83,55 | 91,6 | 91,3 | 97,5 | 91,7 | 94,5 |
| 5 | ERNIE | **85,754** | 90,4 | 74,4 | 97,5 | 93,5 | 91,4 | 92,45 | 93 | 92,6 | 92,8 | 75,2 | 90,9 | 83,05 | 91,4 | 91 | 96,6 | 90,9 | 94,5 |

Table 8: SuperGLUE results with all metrics for the tasks, including tasks with double metrics.

| Rank | Model | GM | AM | BoolQ | CB F1 | CB Acc | CB mean | COPA | Multi RC F1 | Multi RC Acc | Multi RC Mean | ReCoRD F1 | ReCoRD Acc | ReCoRD mean | RTE | WiC | WSC |
|---|---|---|---|---|---|---|---|---|---|---|---|---|---|---|---|---|---|
| 3 | Human | **88,729** | 89,8 | 89 | 95,8 | 98,9 | 97,35 | 100 | 81,8 | 51,9 | 66,85 | 91,7 | 91,3 | 91,5 | 93,6 | 80 | 100 |
| 1 | DeBERTa | **87,601** | 90,3 | 90,4 | 95,7 | 97,6 | 96,65 | 98,4 | 88,2 | 63,7 | 75,95 | 94,5 | 94,1 | 94,3 | 93,2 | 77,5 | 95,9 |
| 2 | T5 + Meena | **87,097** | 90,2 | 91,3 | 95,8 | 97,6 | 96,7 | 97,4 | 88,3 | 63 | 75,65 | 94,2 | 93,5 | 93,85 | 92,7 | 77,9 | 95,9 |
| 4 | T5 | **86,567** | 89,3 | 91,2 | 93,9 | 96,8 | 95,35 | 94,8 | 88,1 | 63,3 | 75,7 | 94,1 | 93,4 | 93,75 | 92,5 | 76,9 | 93,8 |
| 6 | PAI Albert | **85,784** | 86,1 | 88,1 | 92,4 | 96,4 | 94,4 | 91,8 | 84,6 | 54,7 | 69,65 | 89 | 88,3 | 88,65 | 88,8 | 74,1 | 93,2 |
| 5 | NEZHA Plus | **82,294** | 86,7 | 87,8 | 94,4 | 96 | 95,2 | 93,6 | 84,6 | 55,1 | 69,85 | 90,1 | 89,6 | 89,85 | 89,1 | 74,6 | 93,2 |