# OpenReview forum: "How not to Lie with a Benchmark: Rearranging NLP Learderboards"
_NeurIPS.cc/2021/Conference — NeurIPS 2021 Submitted_

### Official Review · Reviewer_hz3e · 2021-06-28

**Rating:** 3
**Confidence:** 4

**Summary:**

The paper proposes to replace the arithmetic mean with harmonic and geometric mean in leaderboard result aggregation. It presents reordered leaderboards for GLUE, SuperGLUE, XGLUE and XTREME.


**Limitations And Societal Impact:**

I was a bit underwhelmed by the lack of discussion of related work, as well as of a more general discussion of leaderboards. Finally, with three different ways of averaging leaderboard results, we enable cherry-picking. The authors could perhaps recommend reporting all three to prevent this.

**Main Review:**

In a paper whose only contribution is to replace arithmetic mean with harmonic and geometric mean, in leaderboard averaging, I would have liked to see the explanation of why these are better than the arithmetic mean, fleshed out in more detail, rather than simply referring to previous work and stating that this is how things are.

The paper fails to discuss how the proposal is related to previous work on aggregation of benchmark results; see [0-3] for a few examples.

That benchmarks use ‘wrong averaging’ is not a new observation. See blog posts such as [4-5]. I feel the paper contributes very little compared to what’s out there; in fact it goes into less detail than some of the blog posts.

The paper comes across as written in haste with poor formatting and grammar, e.g., lines 27-30, and poor language, e.g., "a more balanced balance”. Consider this sentence for example (lines 104-5): "The harmonic mean of the task results as a better overall metric has the same grounding as introduction of the F-1 measure over precision and recall.” I simply cannot make sense of it.

The reasoning is also weak in places. See, for example, lines 79-80: "DecaNLP (McCann et al., 2018) makes a rating using not the average, but the sum of points for all tasks. This approach allows balancing…” I do not understand why summing is easier to de-aggregate than averaging (but let me know if I’m missing something). Another example, line 89: While I agree mean and mode probably do not make sense for aggregation in benchmarks, I don’t understand why "on them such metrics can give the same results for competing systems”.

[0] http://users.umiacs.umd.edu/~jbg//docs/2021_acl_leaderboard.pdf
[1] https://www.aclweb.org/anthology/N13-1068.pdf
[2] https://www.cambridge.org/core/journals/natural-language-engineering/article/survey-of-25-years-of-evaluation/E4330FAEB9202EC490218E3220DDA291
[3] https://www.aclweb.org/anthology/2020.emnlp-main.393/
[4] https://towardsdatascience.com/on-average-youre-using-the-wrong-average-geometric-harmonic-means-in-data-analysis-2a703e21ea0
[5] https://machinelearningmastery.com/arithmetic-geometric-and-harmonic-means-for-machine-learning/

**Time Spent Reviewing:**

1

---

### Official Review · Reviewer_65MJ · 2021-07-13

**Rating:** 2
**Confidence:** 5

**Summary:**

The benchmarks and leaderboards such as GLUE, SuperGLUE have been largely driving the progress of developing NLP models these days. A range of  models have already achieved super-human performance on these leaderboards.  This paper makes an argument that taking the arithmetic mean of scores across different tasks is not ideal, and instead, it proposes to use geometric mean and harmonic mean to re-compute the scores on multiple leaderboards. The findings are 1)  The ranking will be changed greatly by using geometric or harmonic mean; 2) The human performance baseline will rank highest on GLUE and SuperGLUE.

**Limitations And Societal Impact:**

It raises an interesting and important question to the research community for how we compare different models across different metrics/tasks on a leaderboard, although I don't think the paper sufficiently addressed the problem.

**Main Review:**

This paper clearly raises an interesting question and I think how to aggregate different scores/metrics is an important topic that is worth a discussion in the research community. This problem has been somewhat neglected at this point AFAIK. (As the paper pointed out, some other recent benchmarks such as KILT don't use an aggregated score).

However, on the other hand, this paper clearly doesn’t have substantial content for a NeurIPS paper. It is only 6.5 pages and most of the content is re-calculating the scores for 4-5 leaderboards. The contribution is too slim in that regard.

Although the paper proposes to use geometric or harmonic mean, I think it lacks in-depth analysis or explanations for why the metrics should be adopted and used.

All the numbers in the tables have a formatting issue, e.g., “89,80” → “89.80”.

It lacks discussion of related work. The issues of leaderboards have been widely discussed in the NLP community recently, e.g.,
- Utility is in the Eye of the User: A Critique of NLP Leaderboards
- Show your work: Improved reporting of experimental results


**Time Spent Reviewing:**

1

---

### Official Review · Reviewer_Yzmg · 2021-07-17

**Rating:** 3
**Confidence:** 4

**Summary:**

This paper proposes harmonic and geometric mean as methods of aggregating performance results on different tasks in multi-task benchmarks. It then goes through several multi-task benchmarks and shows how the leaderboards would change under these new global metrics.

**Limitations And Societal Impact:**

Yes. I don't think there are any significant concerns. This paper argues for improvements to multi-task metric reporting methodology, which seems like an uncontroversial good (i.e., if we are going to use measuring sticks, we should make them accurate).

**Main Review:**

I really, really like the idea of this paper. It’s a great point that the arithmetic mean has downsides and is probably not entirely appropriate for multi-task benchmarking. It is also heartening that using other aggregation methods can surface new potential for improvement (as it puts humans back on top of the leaderboard). Some theoretical motivation for geometric and harmonic mean measures is provided which is good as well.

I absolutely think this kind of work should be done, but I feel that this particular paper falls short. As is, it seems to me like a missed opportunity to investigate the questions which would be most important for people producing new benchmarks, in particular: which aggregation methods *should* we use, and in what cases? How do different aggregation methods affect the interpretation of the results? And, how can we structure our benchmarks so that we *can* have meaningful aggregate measures over them? The investigation of geometric and harmonic mean with existing multi-task benchmarks provides us an opportunity to really dig into these questions, but I feel that the opportunity is mostly passed up. The theoretical motivation is shallow, just mentioning things like geometric means being appropriate for fractions and harmonic means being appropriate for rates (L120–122). As a minor note, I think a paper like this should go deeper into the background and explain why these things are true instead of just giving references. But more importantly, what I think must be discussed is how these considerations apply to benchmarks in practice. Which common measures in dataset evaluations are meaningfully interpreted as rates versus ratios? How do we deal with benchmarks that have heterogeneous metrics? And for that matter, what even are the desiderata for an aggregate metric in the first place? How do we know when to prefer one over the other, either in general or for any specific benchmark?

So, yeah. Lots of questions remain, and the paper left plenty of space on the table that could have been used to deeply investigate them (I'm not asking for answers; these are hard questions. But at least dedicated attention and investigation which provides the lay of the land is essential for this paper to be really useful). So I feel this work is very promising and interesting, but incomplete.

As a “historical” note which relates to the discussion of the goals of a benchmark/metric: In the background, GLUE is presented as a natural point in the progressive development of benchmarking practice in AI, and arithmetic mean is evaluated from that point of view. But I think it’s important to remember the history here: GLUE was presented particularly as a transfer learning benchmark. The different sizes of test set were a result of the different availability of data for each task, and all tasks were equally weighted partly with the intent of assessing a model’s ability to perform well on tasks with less data by leveraging training signal in other tasks with more data. Other domain- and problem-specific concerns may apply to similar multi-task benchmarks (for example, testing crosslingual performance, etc.) which mean the overall concerns about arithmetic mean might be overblown (rather, the problem would be if the benchmark is interpreted in isolation as a singular barometer of AI performance. But with so many distinct multi-task benchmarks out there these days, I’m not sure if this should be a big concern).

* L36: “a more balanced balanced” -> you can just say “a better balance” :)
* L50–51: the reference to Turing (2009) is a bit strange. His publication “Computing Machinery and Intelligence” appeared in the journal Mind in 1950. Also, it’s not really a good example of an evaluation benchmark. The Turing Test was closer to something between a thought experiment and a grand challenge. Benchmarks are more for measuring incremental progress. A better reference would probably be something on IBM’s Common Task Framework or the Shared Tasks common in the NLP and CV communities.
* L79–80: For the purposes of ranking, sum and average are equivalent, so I’m not sure what this is saying.
* L239: arrengemend

**Time Spent Reviewing:**

1

---

### Official Review · Reviewer_DNNE · 2021-07-19

**Rating:** 4
**Confidence:** 4

**Summary:**

This paper presents an analysis of how results are re-ordered on a number of popular NLP benchmarks when the harmonic or geometric mean are used instead of the arithmetic mean. The paper does not contain much analysis of the significance of these results, or a strong argument in favor of any one approach.


**Limitations And Societal Impact:**

This paper doesn't provide a strong argument for any one method of aggregating sub-task scores, and would be a lot stronger with a detailed discussion of what the different means are highlighting in terms of sub-task performance and expected generalization to new tasks.


**Main Review:**

A number of poular NLP benchmarks aggregate scores from multiple tasks. This paper presents a criticism of the averaging technique, along with an analysis of how results are re-ordered if the harmonic or geometric mean is used instead. The paper shows that the alternate means lead to significantly different orderings and consistently place human scores above NLP systems.

The choice of averaging procedure is important. However, this paper does not present much justification for either of the alternatives. There are citations to related discussions on averaging of rate metrics in reporting computer performance, where outliers can be extremely important. But there is no discussion of the nature and importance of outliers in our NLP benchmarks and this paper does not contains strong evidence for preferring either the harmonic or geometric means.

The paper would be stronger with a thorough analysis of the nature of the outliers that are being highlighted by the harmonic and geometric means, along with a discussion of how the choice affects our abilities to predict performance on arbitrary new tasks. The discussion in this paper is important, but the current presentation does not have enough depth to be presented in NeurIPS.


**Time Spent Reviewing:**

2

---

### Decision · Program_Chairs · 2021-09-27

**Decision:**

Reject

**Comment:**

Reviewers agree that this paper proposes a reasonable small change to the reporting of scores from multitask benchmark, but that the arguments justifying this change are not precise or novel enough to warrant presentation at NeurIPS.